# The Indirect Effect of Democracy on Economic Growth in the MENA Region (1990–2015)

**Shereen Nosier [1,2] and Aya El-Karamani [2,*]**

[1]   Department of Economics, Faculty of Economic Studies & Political Science, Alexandria University, Smouha, Alexandria 21648, Egypt; sheeren.adel@alexu.edu.eg
[2]   Bibliotheca Alexandrina, Seragaldin Institute for Multidisciplinary Advanced Research (SIMAR), Chatby, Alexandria 21526, Egypt; shereen.nosier@bibalex.org
[*]   Correspondence: aya.elkaramani@bibalex.org; Tel.: +20-111-001-6668

**Abstract:** This paper examines the indirect effect of democracy on economic growth using a dataset of 17 MENA countries from 1990 to 2015. Democracy is assumed to affect growth through a series of channels: education, health, physical capital accumulation per labor, government consumption, and trade openness. A system of six simultaneous equations using 3SLS, is used to estimate the effect of democracy on growth through these channels. For further analysis, the countries are classified into groups according to the democratic status on the one side, and the level of income on the other. The results indicate that democracy enhances growth through its positive effect on health in all classifications of countries within the MENA region. However, the effect of democracy on growth through education and physical capital/labor is non-monotonic. Democracy hinders growth through government size and trade openness. Once all of these indirect effects are accounted for, the overall effect of democracy on growth is negative in less democratic countries and poor countries, but positive in more democratic countries and rich countries.

**Keywords:** economic growth; democracy; MENA; simultaneous equations

## 1. Introduction

The direct relationship between democracy and economic growth has been widely investigated in the literature in the last 50 years, and several theories have been developed regarding the relationship between the two variables. Democracy is not just one of the factors affecting economic growth; in fact, it creates the appropriate environment for various other factors to work effectively towards enabling growth. Therefore, even though the direct effect of democracy on growth may not be conclusive or significant, the indirect effect of democracy on growth via different linking channels is highly significant.

The main purpose of this study is to estimate the indirect effect of democracy on economic growth through different channels in the Middle East and North Africa (MENA) region for the period 1990–2015. We estimated a full system of equations determining economic growth and the channel variables using panel simultaneous equations model, specifically three-stage least squares (3SLS). First, we identify the most important channels through which democracy can affect economic growth and estimate the effect of democracy on these identified channels. Second, we estimate the effect of these channels on economic growth in the MENA region. Then, we calculate the indirect effect of democracy on growth. We estimate the indirect effect for different democratic groupings within the region separately to investigate whether the relationship relates to the stage of democratic transition of countries or not. Finally, we estimate the indirect effect of democracy on growth in rich and poor countries within the region separately.

Our results conclude that the overall indirect effect of democracy on economic growth is significantly positive in the more democratic countries but turns negative in the less democratic countries within the MENA region. Therefore, the effect of democracy on economic growth is non-monotonic and varies according to the stage of political transition of countries. Moreover, our results indicate that democracy might induce growth in rich countries, but hinder it in poor countries.

## 2. Economic and Political Background of MENA Countries

### 2.1. The Political Background of MENA Countries

The MENA region is currently well known for the domination of authoritarian regime types whether they are monarchies or national republics where authority is centralized in the hands of a single ruler who is usually persistent for several years or decades, and the absence of representative governance. Sometimes ad-hock elections or small-scale electoral representations are present but mostly, real democracy does not exist within several countries in the MENA region. It seems that the region is well behind the world regarding the process of political transition.

According to the Varieties of Democracy (V-DEM) institute, countries are divided into four categories; liberal democracy, electoral democracy, electoral autocracy, and closed autocracy. In 2016, only Tunisia was classified as a liberal democracy in the MENA region, and Lebanon an electoral democracy. Algeria, Djibouti, Egypt, Iran, Iraq, Sudan, and Turkey are electoral autocracies, whereas Jordan, Kuwait, Libya, Morocco, Oman, Qatar, Saudi Arabia, and Yemen are closed autocracies.

### 2.2. The Economic Background of MENA Countries

In 1990, the real gross domestic product (GDP) of the region amounted to $1626 billion, equivalent to about 4.3% of the world GDP, it increased to $4304 billion, accounting for 5.7% of world GDP in 2015 (World Bank 2016).

The average per capita GDP in the region was $7498 in 2015, more than one and a half that of middle-income countries worldwide. However, gross domestic product per capita (GDPP) in individual MENA countries differs greatly. The region has some of the richest countries in the world; Qatar had the fifth highest GDP per capita in the world in 2015. Other high-income countries include United Arab Emirates, Kuwait, Bahrain, Saudi Arabia, and Oman[1]. The average GDPP of these five countries, together with Qatar, was $32,728 in 2015.

Turkey, Lebanon, Libya, Iran, Iraq and Algeria are considered as upper middle-income countries. The average per capita GDP of this group was about $7096, three times more than the GDPP in lower-middle income countries within the same region. Turkey alone has twice the income per capita of Lebanon, the second highest income in the same group, and three times of GDPP in Algeria, which is the poorest country in that group (World Bank 2016).

On the other hand, the MENA's lower middle-income countries are Syria, Yemen, Djibouti, Sudan, west bank and Gaza, Egypt, Morocco, Jordan and Tunisia. Average GDPP for this group was only $2332 in 2015. Syria, Yemen, Djibouti, and Sudan are considered the poorest countries in the region. Because of internal conflicts and wars, Yemen and Syria have seen a dramatic decline of their GDPP to less than $800 for each (World Bank 2016).

## 3. Literature Review

A new trend in economic growth/democracy literature suggests that the direct effect of democracy on economic growth can be insignificant; however, the indirect effect is significant and prominent.

---

[1] According to the World Bank classification (Atlas method), low-income economies are defined as those countries with a GNI per capita of $1025 or less in 2015. Lower middle-income economies are countries with a Gross National Income (GNI) per capita between $1026 and $4035. Upper middle-income economies are those with a GNI per capita between $4036 and $12,475. Finally, high-income economies are those with a GNI per capita of $12,476 or more (World Bank 2016).

To measure the indirect effect, the total effect of democracy on growth is decomposed into its different components; mainly human capital, physical capital accumulation, income distribution, political stability and population growth, among others (Baum and Lake 2003; Sturm and De Haan 2001; Tavares and Wacziarg 2001).

Baum and Lake (2003) used a sample of 128 countries worldwide over the period from 1967 to 1997, and concluded that democracy has no statistically significant direct effect on growth. However, there are significant indirect effects of democracy on growth through increased life expectancy in poor countries and increased secondary education in non-poor countries. Tavares and Wacziarg (2001) applied their model on a panel of 65 industrial and developing countries from 1970 to 1989 and concluded that democracy fosters growth by improving the accumulation of human capital and lowering income inequality. On the other hand, democracy hinders growth by reducing the rate of physical capital accumulation and less robustly, by raising the ratio of government consumption to GDP. When summing up the indirect effects of democracy on growth, the negative effect through physical investment dominates and the overall effect of democracy on economic growth is moderately negative.

Fabro and Aixalá (2012) used simultaneous equations model—weighted Two-Stage Least Squares (2SLS)—over the period from 1976 to 2005 to estimate the direct and indirect effects of economic and political freedom on the economic growth of 79 countries worldwide. The results show that democracy, represented in civil liberties, political freedom and economic freedom, is important for economic growth either through a better allocation of resources or, indirectly, through the stimulation of investment in physical and human capital. Helliwell (1994) used the instrumental variables technique to examine empirically the linkages between democracy and economic growth in 90 countries worldwide over the period from 1960 to1985. He concluded that democracy has a negative, but statistically insignificant direct effect on growth, but a larger positive indirect effect through education and investment. Moreover, by applying a simultaneous equations model, 3SLS, on a sample of 96 developed and developing countries worldwide, Feng (1997) aimed to investigate the relationship between democracy, political stability and economic growth over the period from 1960 to 1980. The results indicated that democracy has a positive indirect effect upon growth through its impacts on the probabilities of both regime change and constitutional government change from one ruling party to another. Finally, Przeworski and Limongi (1993) investigated the effect of political regimes on economic growth. They indicated that political regime has no direct effect on growth, but it increases GDP per capita through its negative effect on population growth rate.

A few studies have argued that the effect of democracy on growth could be non-linear, differing according to the stage of democracy, or the initial standards of living of the countries under consideration (Kuznets 1955; Barro 1996). Barro (1996) used a sample of 100 countries worldwide to analyze the effect of democracy on economic growth. He assessed the direct relationship between democracy and economic growth, while the initial GDP per capita, education, health, population growth rate, government size, black-market-premium, and rule-of-law variables were held constant to isolate the direct effect of democracy on growth. The overall effect of democracy on growth turned out to be weakly negative. Upon omitting the rule of law, education, health and fertility rate variables from the model, the effect of democracy on growth became positive and significant. Barro attributed the favorable effect that democracy had on growth to a positive correlation between democracy and these omitted variables that are themselves growth promoting. Upon replacing the democracy index with two dummy variables representing democracy index score, Barro found evidence of a non-linear relationship between democracy and growth. The results reported that the middle level of democracy is the most favorable for growth, while the lowest and highest levels had lower growth rates. Moreover, the political Kuznets hypothesis (Kuznets 1955) states that at the first stage of the political democracy, democracy redistributes income negatively because of its negative effect on income equality, thereby reducing economic growth. However, in the long run, democracy reduces income inequality and thus supports economic growth. Therefore, the relationship between democracy and economic growth is non-monotonic.

A limited number of papers illustrated the direct effect of democracy on economic growth in African and MENA countries. None of them investigated the indirect effect of democracy on growth via different channels. Zghidi (2017) used a panel of 31 African countries over the period from 1986 to 2014 to investigate whether political stability and democracy increase the economic growth rate, using Generalized Method of Moment (GMM) dynamic panel data analysis. She concluded that both democracy and political stability have a positive and significant effect on economic growth. Rachdi and Saidi (2015) used a sample of 17 MENA countries to investigate empirically the effect of democracy on growth during the period from 1983 to 2012, using two different models: a Fixed Effect (FE) and Random Effect (RE) regression model, and a GMM model. They repeated the test five times for each model using several measures of democracy that capture different aspects of democracy and found the effect of democracy to be negative and statistically significant on economic growth for both models for four out of the five measures of democracy.

This work examines the indirect effect of democracy on growth through various channels. The study attempts to classify the estimated models into homogenous groups according to the stage of democratic transition of the groups of countries.

## 4. Model Specification

We specified a panel data model to estimate economic growth in 17 MENA countries over the period from 1990 to 2015. Education, health, physical capital per labor, government size, and trade openness were identified as important channels to include in our model. Therefore, the model includes six equations, an economic growth equation, as well as the five channel equations.

The estimated democracy coefficients of each equation are sensitive to the chosen specification, and especially to the exclusion of particular endogenous or exogenous variables. For the growth equation, the specification is derived from an augmented Solow model, with the set of channel variables as independent variables. The equation is expressed as follows:

$$lnGDPP_{it} = \gamma_0 + \gamma_1 lnMR_{it} + \gamma_2 lnEDUS_{it} + \gamma_3 lnGCFL_{it} + \gamma_4 lnGZ_{it} + \gamma_5 lnTR_{it} + u_{it} \tag{1}$$

where $t = 1, 2, \ldots, 26$ years (1990–2015), $i = 1, 2, \ldots, 17$ MENA countries. $lnGDPP_{it}$ is the natural log of per capita income in period $t$, at each country $i$, $lnMR_{it}$ is the natural log of mortality rate, $lnEDUS_{it}$ is the natural log of secondary school enrollment rate, $lnGCFL_{it}$ is the natural log of gross capital formation per labor, $lnGZ_{it}$ is the natural log of government size, $lnTR_{it}$ is the natural log of trade openness, and $u_{it}$ residuals. $\gamma_0, \gamma_1, \gamma_2, \gamma_3, \gamma_4, \gamma_5$ are the parameters to be estimated, and according to the theory, it is expected that: $\gamma_1 > 0, \gamma_2 > 0, \gamma_3 > 0, \gamma_4 > 0, \gamma_5 > 0$.

The democracy index as well as income per capita are included in all the channel equations of the model. In this paper, we measure democracy by the electoral democracy index ($EDEM_{it}$) obtained from the V-DEM institute, University of Gothenburg (Coppedge et al. 2016). The health equation is represented by infant mortality rate ($lnMR_{it}$). Female education ($lnEDUPF_{it}$) is considered an important determinant of health in the MENA region. Primary school education ($lnEDUP_{it}$), health ($lnMR_{it}$), among other variables are the determinants of the education equation, which is the second channel equation.

$$lnMR_{it} = \beta_0 + \beta_1 EDEM_{it} + \beta_2 lnGDPP_{it} + \beta_3 lnEDUPF_{it} + u_{it} \tag{2}$$

$$lnEDUS_{it} = \beta_0 + \beta_1 EDEM_{it} + \beta_2 lnGDPP_{it} + \beta_3 lnEDUP_{it} + \beta_4 lnMR_{it} + u_{it} \tag{3}$$

$$lnGCFL_{it} = \beta_0 + \beta_1 EDEM_{it} + \beta_2 lnGDP_{it} + \beta_3 lnEDUS_{it} + \beta_4 lnEX_{it} + \beta_5 lnUPOP_{it} + \beta_6 lnUM_{it} + u_{it} \tag{4}$$

$$lnGZ_{it} = \beta_0 + \beta_1 EDEM_{it} + \beta_2 lnGDP_{it} + \beta_3 lnCO_{it} + \beta_4 lnPOPG_{it} + \beta_5 lnTR_{it} + u_{it} \tag{5}$$

$$lnTR_{it} = \beta_0 + \beta_1 EDEM_{it} + \beta_2 lnGDP_{it} + \beta_3 lnLA_i + \beta_4 lnUPOP_{it} + \beta_5 INF_{it} + u_{it} \tag{6}$$

Physical investment per labor ($lnGCFL_{it}$) is the third channel, expressed by the gross capital formation per labor. Education, exchange rate, urban population and unemployment rate represent the main explanatory variables in equation 4. Equation 5 represents the government size channel, the government final consumption expenditure (% GDP) is an appropriate measure for this variable in the literature ($lnGZ_{it}$). Corruption index ($lnCO_{it}$), population growth ($lnPOPG_{it}$), trade openness ($lnTR_{it}$) are important regressors in this equation. Finally, the openness equation includes the country area ($lnLA_i$), inflation ($INF_{it}$) and urban population ratio ($lnUPOP_{it}$) as important determinants.

Data for all the variables, except democracy index is transformed into natural logarithm, therefore the estimated parameters in this form are the elasticities and the difference in logs approximate the growth rates, so the results are interesting and easy to interpret. The data, and its sources and measures, are represented in Appendix 7, Table A1.

The previous specification is applied on various models, to test the sensitivity of our results on the one hand and to answer the following two questions on the other. Does this indirect effect of democracy on growth differ according to the state of political transition of each group of countries within the MENA region? Does this indirect effect of democracy on growth depend on the level of income per capita of these countries? To achieve the main aim of this study we classified countries within the MENA region into two groups, relatively democratic countries, which have achieved a minimum degree of democratic transition, and countries that are relatively autocratic within the region. We use, first, the classification of Freedom House of 'free and partly free' countries as one model and 'not free' countries as the other model. Second, we use the V-DEM institute classification of countries as 'electoral and liberal democracy and electoral autocracy' countries group as one model and 'closed autocracy' countries as the other model. Moreover, we classify the MENA countries into democratic countries and autocratic countries based on the average electoral democracy index score over the study period.

We then created another classification according to GDP per capita by dividing the countries into rich countries, which have an income of more than $4000 per capita annually, and poor countries, which have less than $4000 per capita annually. This number was decided as the average of the per capita GDP of all the 17 countries in the sample over the period of the study. This is equivalent to the World Bank classification, illustrated in Section 2.2, since we consider low income and lower-middle income countries as poor, but upper-middle income and high-income economies as rich. To capture the different effects of democracy on growth in poor and rich countries, if any, we included a dummy variable in our different channel equations. This dummy was coded 1 if the country is poor, and 0 otherwise. We then separately interacted this variable with democracy. The resulting interaction terms, democracy in poor countries and democracy in rich countries are included in all of our channel equations to capture the quantitatively distinct effects of democracy on growth via different channels in poor and rich countries. Details of different groups of countries in each model and descriptive statistics for variables in each model are illustrated in Appendix 7, Tables A2–A8.

## 5. Methodology

An important assumption of ordinary least squares (OLS) estimators is that the regressors are exogenous. If this assumption is violated, regressors are correlated with the error term and OLS are biased and inconsistent. To solve this problem, simultaneous equations model using 2SLS estimator can be utilized instead since it is consistent even if the explanatory variables are endogenous. However, if the errors terms of various equations are correlated, the 2SLS estimators are consistent but inefficient. Seemingly Unrelated Regression (SUR) is efficient because it takes into account the correlation of errors across equations. However, it assumes that there are no endogenous variables on the right-hand side. If endogenous variables appear on the right-hand side of equations, the system estimation of SUR must be combined with the instrumental variables method of 2SLS. The resulting estimator is the 3SLS. While both 2SLS and 3SLS are consistent, the 3SLS is asymptotically more efficient than 2SLS because

it uses information on the correlation of the stochastic disturbance terms of the structural equations (Intriligator 1978).

The 3SLS estimation contains three stages. First, all reduced-form coefficients are estimated applying the least square estimator. Second, structural coefficients are estimated, using 2SLS to each of the structural equations. Third, all of the structural coefficients of the system are estimated using Generalized Least Squares (GLS) estimators, using a covariance matrix for the stochastic disturbance terms of the structural equations, which is estimated from the second-stage residuals. Using the information contained in this covariance matrix improves efficiency (Intriligator 1978).

To construct our model, we estimated a full set of six simultaneous equations with six dependent variables, using 3SLS estimators—Following (Tavares and Wacziarg 2001) and (Baum and Lake 2003). The estimated coefficients of the economic growth equation $(\hat{\gamma}_s)$—Equation (1)—yield the effect of the channels on the growth in MENA region, whereas the estimated coefficients of the channel equations $(\hat{\beta}_s)$—Equations (2)–(6)—represent the effect of democracy on the channels. The product of the coefficient of democracy in the channel equation by the coefficient of the channel variable in growth equation illustrates how democracy affects growth indirectly through this particular channel. The summation of these calculated coefficients yields the total indirect effect of democracy on growth in the MENA region.

As mentioned above, the indirect effects of democracy on growth via each channel variable are calculated by multiplying coefficients across equations. Therefore, the statistical significance of these coefficients $(\hat{\delta}_s)$ is not straightforward. The delta method (Oehlert 1992) is utilized to calculate the standard errors of these calculated coefficients, assuming that the covariance between $\hat{\gamma}$ and $\hat{\beta}$ is zero, the standard errors of the indirect effect of democracy on growth can be calculated as in Equation (7).

$$\text{SE}(\hat{\delta}_i) \;=\; \sqrt{\hat{\gamma}_i^2 \text{SE}(\hat{\beta}_{1i}^2) + \hat{\beta}_{1i}^2 \text{SE}(\hat{\gamma}_i^2)} \tag{7}$$

where $\hat{\delta}_i$ is the calculated indirect effect of democracy on economic growth via each channel variable, i denotes each channel variable, education, health, physical capital/labor, government size, and trade openness respectively. $\hat{\gamma}_i$ is the estimated coefficient of each channel variable in Equation (1), $\hat{\beta}_{1i}$ is the estimated coefficients of democracy in each channel equation (Equations (2)–(6)). SE is the standard errors of the estimated coefficients.

## 6. Empirical Results

First, the effect of the different channels on economic growth is displayed in Section 6.1. Then, the effect of democracy on each channel is represented in Section 6.2. Lastly, the indirect effect of democracy on growth via each channel is calculated in different models based on the stage of democracy in each group of countries within the region, as illustrated in Section 6.3. Section 6.4 displays the estimated indirect effect coefficients of democracy on growth according to the country's standard of living. Finally, Section 6.5 examines the sensitivity of our results to some modifications of the model.

Before displaying the estimated coefficients of our models, diagnostic tests were performed to check the consistency and efficiency of our estimates. First, Hausman test was applied, as illustrated in Appendix 7, Table A9, and the results confirmed the endogeneity of our models, hence the use of an endogenous technique such as 3SLS. Moreover, the Variance Inflation Factor (VIF) was applied to our estimated models—after performing 2SLS—and confirmed the absence of multicollinearity in all of our estimated equations, as illustrated in Appendix 7, Table A10[2].

---

[2]　Only in two cases out of 36, the VIF coefficients exceed 10. This is in Equation (2) (where education is the dependent variable) in the closed autocracy model and in Equation (6) (where trade is the dependent variable) in the free and partly free model.

*6.1. The Effect of Different Channels on Economic Growth (1990–2015)*

Table 1 illustrates the results of estimating Equation (1), which represent the effect of each channel variable on economic growth. The estimated parameters are highly significant and consistent with the economic theory. The effect of education and health on growth is very strong and positive in all the models. This is expected, as human capital improvement enhances productivity, and consequently, increases economic growth. In the more democratic countries[3], the effect of education is stronger than its effect in less democratic ones. A 1% increase in secondary school enrolment leads to a 0.55% increase in economic growth on average in democratic groups, but only 0.28% increase in growth in the less democratic ones. As asserted by many authors (Barro 1996; Freund and Jaud 2014) democratic countries reduce education inequality, which maximizes the effect of education on growth. On the contrary, the effect of health on economic growth is stronger in less democratic countries than in more democratic ones. A 1% decrease in mortality rate leads to an increase in growth by 0.53% and 0.96% on average in more and less democratic groups, respectively.

**Table 1.** The Effect of the Channels on Economic Growth in Different Models.

| Effect of Channel on Growth | Education | Health | Ph. Capital | Gov. Size | Trade | $R^2$ |
|---|---|---|---|---|---|---|
| **Free and partly free** | 0.575 [4.53] | −0.301 [−3.32] | 1.111 [20.52] | 0.893 [5.82] | −0.282 [−2.94] | 0.89 |
| **Not free** | 0.269 [4.19] | −0.935 [−16.44] | 0.833 [27.19] | 0.505 [6.76] | −0.224 [−6.47] | 0.86 |
| **Electoral and liberal democracy and electoral autocracy** | 0.550 [9.39] | −0.689 [−14.14] | 0.514 [15.21] | 0.531 [6.25] | −0.215 [−6.75] | 0.79 |
| **Closed autocracy** | 0.285 [2.71] | −1.179 [−15.63] | 1.086 [29.00] | 0.209 [2.03] | −0.427 [−3.70] | 0.90 |
| **Democratic countries** | 0.525 [6.99] | −0.599 [−9.54] | 0.723 [18.93] | 0.595 [5.90] | −0.253 [−7.39] | 0.85 |
| **Autocratic countries** | 0.272 [2.57] | −0.762 [−9.62] | 0.955 [27.11] | 0.546 [6.55] | −0.246 [−3.01] | 0.88 |

T-statistics are included in Parentheses.

The increase in physical capital formation per labor boosts the economy in all models; a 1% increase in investment per labor induces the growth by 0.87% on average of all models. Physical capital formation increases production and creates more employment opportunities. Furthermore, it leads to technical progress, which helps in achieving economies of large-scale production, increases specialization and provides machines and equipment for the growing labor force. Therefore, it leads to the expansion of the market (Shuaib et al. 2015). Consequently, the increase in physical capital formation is expected to foster the economy strongly. On the other hand, trade openness in the MENA region hinders economic growth in all models, with an elasticity of 0.30 on average. This could be attributed to the fact that the imports of these countries mostly exceed their exports, and that the terms of trade tend to be in disfavor of them. There is no significant difference between the effect of these two variables on growth in democratic and autocratic groups of countries.

---

3　Authors mean by "more democratic" Free and partly free (from the freedom house classification), Electoral and liberal democracy and electoral autocracy (from V-DEM classification) and Democratic-countries (based on data classification). "Less democratic" means not free (from the freedom house classification), Closed autocracy (from V-DEM classification), and Autocratic-countries (based on data classification).

Government size enhances economic growth, since more government expenditure on infrastructure, education and health affects economic growth positively. In addition, the effect of government size on growth is stronger in democratic countries than in autocratic countries, with an elasticity ranging from 0.42 on average for the relatively autocratic groups to 0.67 on average for the relatively democratic groups of countries.

*6.2. The Effect of Democracy on the Link Variables (1990–2015)*

Table 2 illustrates the elasticity of the different channels with respect to democracy, represented by the coefficient $(\hat{\beta}_{1s})$ in Equation (2) to Equation (6), in different models. These coefficients represent the effect of democracy on each channel variable. The effect of democracy on education is significant and positive in the democratic countries, indicating that a 1% increase in democracy index tends to increase education by 0.22% on average. However, the same effect turns negative in the autocratic groups, with a marginal effect that equals −0.65 on average. The effect of democracy on mortality rate is negative and strong in all models. A 1% increase in democracy tends to decrease mortality rate by 1% on average in all models. This is explained by the fact that democracies tend to be usually more responsive to the basic needs of the people than dictatorships. They will choose policies that promote human capital accumulation, even if on account of physical capital.

**Table 2.** The Effect of Democracy on the Channels in Different Models.

| Effect of Democracy on the Channel | Education | Health | Ph. Capital | Gov. Size | Trade |
|---|---|---|---|---|---|
| **Free and partly free** | 0.239 [3.33] | −1.029 [−3.14] | 0.499 [3.75] | −0.573 [−2.89] | 1.003 [5.16] |
| **Not free** | −0.669 [−5.82] | −0.317 [−2.02] | −0.971 [−3.17] | 0.969 [3.97] | 2.199 [3.89] |
| **Electoral and liberal democracy and electoral autocracy** | 0.166 [1.49] | −0.830 [−5.10] | 1.258 [4.65] | −0.779 [−3.02] | 3.019 [4.84] |
| **Closed autocracy** | −0.752 [−5.51] | −1.041 [−6.69] | −0.865 [−2.67] | −0.383 [−2.07] | 0.557 [2.61] |
| **Democratic Countries** | 0.250 [2.91] | −0.868 [−3.73] | 0.636 [2.71] | −0.585 [−4.43] | 2.543 [4.86] |
| **Autocratic Countries** | −0.533 [−5.42] | −2.448 [−9.98] | −1.787 [−4.64] | −1.454 [−4.62] | 1.126 [2.60] |

T-statistics are included in Parentheses.

The effect of democracy on physical investment per labor is also positive in democratic countries (0.80 on average), it turns negative in less democratic countries, equal to −1.20 on average. Democracy could affect physical capital formation positively or negatively. Several researchers argue that physical investment grows in a climate of liberty, free-flowing information and property rights. Democracy reduces the extent of political, social and economic uncertainty, and in turn encourages physical capital formation. On the contrary, other researchers claim that democracy may redistribute national income in favor of labor and disfavor of capital, by giving a greater voice to unions and labor wages and interests. Higher wages increase the cost of production, decrease the profits, and thus lower the incentives for private investment (Tavares and Wacziarg 2001).

The effect of democracy on government size is always negative and significant, a 1% increase in democracy decreases government size by 0.47% on average. More likely autocrats intend to increase the size of government to maximize their influence and control over the economy since their power is derived from the resources under their control (Tavares and Wacziarg 2001). On the contrary, democracies tend to decrease government size.

The effect of democracy on trade openness is very strong and positive in all models; a 1% increase in democracy tends to increase trade openness by 1.70% on average for all models. Democracies generally increase economic freedom and benefit a great number of consumers at the expense of a few producers, who receive more advantages from the protectionist policies. Therefore, democracy stimulates economic freedom and trade openness.

To conclude, the effect of democracy on mortality rate and government size is always negative, whereas its effect on trade openness is always positive. The effect of democracy on education and physical capital per labor is non-monotonic; it is negative in less democratic groups of countries but turns positive in the more democratic groups of countries.

*6.3. The Indirect Effect of Democracy on Economic Growth (1990–2015)*

The product of the coefficient of democracy in the channel equation $((\hat{\beta}_{1s})$ in Equations (2)–(6)) by the coefficient of the channel variable $(\hat{\gamma}_s)$ in the growth equation (Equation (1)) illustrates how democracy affects growth indirectly through this particular channel. The overall indirect effect of democracy on economic growth is significant in the MENA region through the effect of all the selected channels, as illustrated in Table 3. The effect of democracy on economic growth is obtained through the effect of health, education and physical capital accumulation per labor. Although the effect of democracy on economic growth via education is unexpectedly negative in autocratic countries, it turns positive when the country is democratic. The latter result is attributed to the effect of democracy on education, as illustrated in Section 6.2. The effect of democracy on economic growth via health is very strong and positive in all the models. It is greater in the less democratic countries than in the more democratic ones. This result is attributed to the effect of health on economic growth as illustrated in Section 6.1.

**Table 3.** The Indirect Effect of Democracy on Economic Growth in MENA Countries.

| DEM/EG (%) | Total | Education | Health | Ph. Capital | Gov. Size | Trade |
|---|---|---|---|---|---|---|
| **Free and partly free** | 0.207 | 0.138 [2.75] | 0.310 [2.28] | 0.554 [3.60] | −0.512 [−2.77] | −0.283 [−0.93] |
| **Not free** | −0.693 | −0.177 [−3.40] | 0.296 [2.00] | −0.790 [−3.15] | 0.489 [3.42] | −0.493 [−3.33] |
| **Elec. and liberal Dem and elec. autocracy** | 0.244 | 0.089 [1.36] | 0.572 [4.80] | 0.647 [4.45] | −0.413 [−2.72] | −0.650 [−3.94] |
| **closed autocracy** | −0.244 | −0.215 [−2.43] | 1.227 [6.15] | −0.939 [−2.66] | −0.080 [−1.33] | −0.237 [−12.63] |
| **Democratic Countries** | 0.120 | 0.131 [2.69] | 0.521 [1.00] | 0.460 [2.68] | −0.348 [−3.54] | −0.644 [−10.81] |
| **Autocratic Countries** | −1.413 | −0.145 [−2.32] | 1.864 [6.93] | −1.707 [−4.57] | −0.793 [−3.13] | −0.632 [−2.53] |

T-statistics of the indirect effect coefficients are calculated using the delta method.

Moreover, the effect of democracy on economic growth through physical capital accumulation per labor is significantly positive in the democratic countries, while it is significantly negative in the autocratic groups. An increase in the democracy index by 1% fosters growth in the democratic groups by 0.55% on average, and hinders growth in the autocratic groups by 1% on average. In democratic countries, where there is a climate of liberty, free flowing information, and secured property rights, the increase in democracy tends to enhance growth. On the other hand, where there is no good institutional framework through which democracy could be practiced; higher levels of democracy give a greater voice to unions, labor wages and sectarian interests. Therefore, the cost of production will increase, and profits will decrease, thereby reducing private investment.

The effect of democracy on economic growth through both government size and trade openness is mostly negative in all models with different specifications. An increase in democracy by 1% decreases growth by 0.43% on average via government size, and by 0.49% on average via trade openness.

From the above discussion, it is clear that the relationship between democracy and growth is non-monotonic via physical capital per labor and education, but it is monotonic via health, government size and trade openness. Therefore, the overall indirect effect of democracy on growth in the MENA region is non-monotonic as illustrated in Figure 1. It differs according to the stage of democratic transition of the group of countries. Democracy fosters growth in the more democratic groups; free and partly free countries, electoral and liberal democracies, and democratic countries according to the electoral democracy index, and hinders growth in the less democratic groups; not free countries, closed autocracies, and the autocratic countries according to the electoral democracy index, within the region.

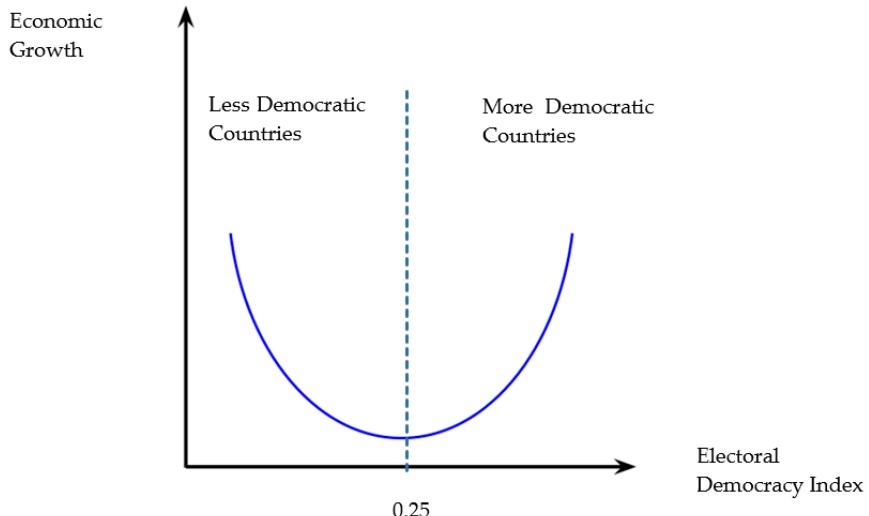

**Figure 1.** The Indirect Effect of Democracy on Economic Growth in MENA Countries.

Our results are in line with the Kuznets hypothesis (Kuznets 1955), which indicates that at the beginning of the political process, democracy decreases economic growth, but after democracy reaches a more mature stage, it enhances economic growth. In addition, our results are consistent with Barro's findings (Barro 1996). The latter's results indicate that democracy has a more favorable effect on growth at a middle level but negative effect in countries with a very low level of democracy. Our results report a negative effect of democracy in countries that are classified as "not free", "closed autocracy", or "autocratic", which usually score very low on the V-DEM electoral democracy index (0.25 or below on average). In addition, democracy has a positive effect on countries classified as "partly-free and free", "electoral democracies", and "democratic groups", which have a democratic index score ranging between 0.3 and 0.55 on average, which falls within the middle range of democracy defined by Barro.

*6.4. The Indirect Effect of Democracy on Economic Growth in Poor and Rich Countries within MENA Region (1990–2015)*

Democracy affects economic growth differently according to the level of per capita income in MENA countries. We investigated this hypothesis by estimating the effect of democracy on the different channels, as illustrated in Table 4. The effect of democracy on education is not affected by the level of income per capita; but it is significantly affected by the democratic stage of the group of countries. However, the effect of democracy on health is much stronger in rich countries than in poor countries. The latter effect is highly significant in all rich countries whatever the stage of democracy in the group, whereas it is insignificant in three out of six poor groups of countries within the region. The economic conditions are very important and mostly a pre-requisite for democracy to affect health significantly.

**Table 4.** The Effect of Democracy on the Channels in Poor and Rich Countries.

| Effect of Democracy on the Channel | Education | Health | Ph. Capital | Gov. Size | Trade |
|---|---|---|---|---|---|
| **Free and Partly Free** | 0.239 [3.33] | −1.029 [−3.14] | 0.499 [3.75] | −0.573 [−2.89] | 1.003 [5.16] |
| **Poor Countries** | 0.297 [2.75] | −0.219 [−0.56] | −0.167 [−1.02] | −0.219 [−0.53] | 0.705 [3.34] |
| **Rich Countries** | 0.227 [3.33] | −1.167 [−3.73] | 0.607 [5.15] | −0.497 [−2.03] | 1.228 [3.10] |
| **Not Free** | −0.669 [−5.82] | −0.317 [−2.02] | −0.971 [−3.17] | 0.969 [3.97] | 2.199 [3.89] |
| **Poor Countries** | −0.644 [−5.62] | −0.291 [−1.89] | −1.162 [−3.14] | 1.038 [4.35] | 2.234 [3.97] |
| **Rich Countries** | −0.732 [−3.94] | −0.800 [−3.14] | 1.464 [3.06] | −0.312 [−0.78] | 0.040 [0.04] |
| **Electoral and Liberal Democracy and Electoral Autocracy** | 0.166 [1.49] | −0.830 [−5.10] | 1.258 [4.65] | −0.779 [−3.02] | 3.019 [4.84] |
| **Poor Countries** | 0.162 [1.42] | −0.460 [−2.24] | 0.141 [0.45] | −0.632 [−2.36] | 2.638 [4.17] |
| **Rich Countries** | 0.546 [3.73] | −0.931 [−5.85] | 2.224 [8.45] | −0.777 [−2.93] | 2.552 [3.10] |
| **Closed Autocracy** | −0.752 [−5.51] | −1.041 [−6.69] | −0.865 [−2.67] | −0.383 [−2.07] | 0.557 [2.61] |
| **Poor Countries** | −1.501 [−6.07] | −0.741 [−2.83] | −6.378 [−14.78] | 1.989 [3.86] | 0.614 [1.82] |
| **Rich Countries** | −0.254 [−1.58] | −0.840 [−4.73] | −0.638 [−1.16] | −0.644 [−2.61] | 0.584 [2.59] |
| **Democratic Countries** | 0.250 [2.91] | −0.868 [−3.73] | 0.636 [2.71] | −0.585 [−4.43] | 2.543 [4.86] |
| **Poor Countries** | 0.386 [3.46] | 0.141 [0.57] | 0.286 [0.94] | −0.424 [−2.97] | 2.583 [3.95] |
| **Rich Countries** | 0.228 [2.65] | −1.066 [−5.11] | 0.692 [2.94] | −0.662 [−5.00] | 2.390 [4.11] |
| **Autocratic Countries** | −0.533 [−5.42] | −2.448 [−9.98] | −1.787 [−4.64] | −1.454 [−4.62] | 1.126 [2.60] |
| **Poor Countries** | −0.358 [−3.51] | −2.844 [−10.71] | −2.343 [−6.28] | −0.442 [−1.19] | 2.295 [4.90] |
| **Rich Countries** | −0.619 [−4.83] | −2.040 [−8.27] | 0.135 [0.31] | −1.740 [−5.79] | −0.288 [−0.72] |

T-statistics are included in Parentheses.

The effect of democracy on physical capital per labor is highly affected by the level of income per capita. Even if the overall effect of democracy on investment is negative in less democratic groups, it turns positive in rich countries. When this effect is positive in the more democratic countries, it is highly significant and strong in rich countries, but turns insignificant in poor countries.

The effect of democracy on government size is mostly negative in rich as well as in poor counties. This effect is stronger—more negative—in rich countries than in poor countries. Moreover, the effect of democracy on trade openness is always positive no matter in poor or rich countries.

Turning to the overall indirect effect of democracy on economic growth, as illustrated in Table 5, important points arise. For all the models, democracy has an overall negative effect on economic growth in poor countries but a positive effect in rich countries. A 1% increase in democracy index fosters economic growth by 0.69% on average in rich countries but hinders growth by 1.03% in poor countries. Our findings are in line with the Lipset (1959) hypothesis, which states that development is a prerequisite for democracy, or that democracy cannot survive in poor and uneducated societies. Poor people do not have the luxury of choosing their rulers and having a voice in the political process. They are too preoccupied with earning their livings.

**Table 5.** The Indirect Effect of Democracy on Economic Growth in Rich and Poor Countries.

| DEM/EG (%) | Total | Education | Health | Ph. Capital | Gov. Size | Trade |
|---|---|---|---|---|---|---|
| **Free and Partly Free** | 0.207 | 0.138 [2.75] | 0.310 [2.28] | 0.554 [3.60] | −0.512 [−2.77] | −0.283 [−0.93] |
| **Poor Countries** | −0.297 | 0.172 [2.34] | 0.063 [0.54] | −0.187 [−1.02] | −0.190 [−0.53] | −0.156 [−1.90] |
| **Rich Countries** | 0.442 | 0.132 [2.66] | 0.337 [2.38] | 0.677 [4.99] | −0.431 [−1.91] | −0.272 [−1.85] |
| **Not Free** | −0.693 | −0.177 [−3.40] | 0.296 [2.00] | −0.790 [−3.15] | 0.489 [3.42] | −0.493 [−3.33] |
| **Poor Countries** | −0.798 | −0.178 [−3.41] | 0.274 [1.88] | −0.954 [−3.88] | 0.586 [3.75] | −0.526 [−3.41] |
| **Rich Countries** | 1.568 | −0.203 [−2.90] | 0.754 [3.08] | 1.202 [3.04] | −0.176 [−0.77] | −0.009 [−0.04] |
| **Elec. and Liberal DEM and Elec. Autocracy** | 0.244 | 0.089 [1.36] | 0.572 [4.80] | 0.647 [4.45] | −0.413 [−2.72] | −0.650 [−3.94] |
| **Poor Countries** | −0.445 | 0.088 [1.41] | 0.331 [2.21] | 0.075 [0.45] | −0.352 [−2.22] | −0.586 [−3.57] |
| **Rich Countries** | 1.155 | 0.296 [3.46] | 0.669 [5.44] | 1.190 [7.45] | −0.433 [−2.67] | −0.567 [−2.83] |
| **Closed Autocracy** | −0.244 | −0.215 [−2.43] | 1.227 [6.15] | −0.939 [−2.66] | −0.080 [−1.33] | −0.237 [−12.63] |
| **Poor Countries** | −3.045 | −0.581 [−3.92] | 0.864 [2.32] | −3.396 [−7.16] | 0.608 [1.78] | −0.256 [−1.64] |
| **Rich Countries** | 0.108 | −0.073 [−1.12] | 0.979 [3.79] | −0.368 [−0.62] | −0.171 [−1.84] | −0.259 [−2.03] |
| **Democratic Countries** | 0.120 | 0.131 [2.69] | 0.521 [1.00] | 0.460 [2.69] | −0.348 [−3.54] | −0.644 [−10.81] |
| **Poor Countries** | −0.636 | 0.200 [3.08] | −0.093 [−0.55] | 0.217 [0.94] | −0.269 [−2.67] | −0.692 [−3.51] |
| **Rich Countries** | 0.282 | 0.118 [2.47] | 0.698 [4.49] | 0.526 [2.91] | −0.420 [−3.88] | −0.640 [−3.62] |
| **Autocratic Countries** | −1.413 | −0.145 [−2.33] | 1.864 [6.93] | −1.707 [−4.57] | −0.793 [−3.13] | −0.632 [−2.53] |
| **Poor Countries** | −0.969 | −0.127 [−2.49] | 2.143 [7.24] | −2.196 [−6.12] | −0.240 [−1.18] | −0.549 [−2.57] |
| **Rich Countries** | 0.567 | −0.219 [−2.86] | 1.537 [6.63] | 0.127 [0.31] | −0.947 [−4.44] | 0.069 [0.70] |

T-statistics are included in Parentheses.

*6.5. Sensitivity Analysis*

The sensitivity to sample coverage, obtained from estimating six different models using the same equations' specification and the same methodology but different samples are illustrated in Sections 6.1–6.3. Three different democratic groups of countries based on different specifications have been estimated, and the results are consistent and follow the same trend in the three different estimated models; free and partly free countries, liberal democracies and electoral autocracies, and democratic countries according to the V-DEM electoral democracy index. In addition, the results of the autocratic groups of countries also follow the same pattern in the three different estimated models, not free countries, closed autocracies, and autocratic countries according to the V-DEM electoral democracy index. Moreover, in this subsection we examine the sensitivity of our results to the time period and to alternative specifications of the economic growth variable.

Firstly, we examined the sensitivity of the estimated coefficients to the period of time, therefore, we estimated our six equations over the period from 1995 to 2015. The estimated coefficients are consistent and stable. However, the coefficient of democracy in the trade openness equation takes a lower value in all of the models in the new sample, which has less degrees of freedom, as illustrated in Appendix 7, Tables A11 and A12.

Secondly, we estimated the effect of democracy on the channel variables in alternative specifications of the economic growth variable, specifically GDP, GDPG, and GDPPG. The estimated coefficients have not changed in different specifications, indicating that the estimated coefficients are not biased. However, although the effect of democracy on health in the health equation still follows the same pattern, it represents relatively higher or lower values than before, as illustrated in Appendix 7, Tables A13–A15.

Finally, the effects of the channel variables on economic growth and the effect of democracy on the channel variables are estimated, and then the indirect effect of democracy on economic growth is calculated using three different estimators; 2SLS, SUR, and OLS for all of our six models. The results are illustrated in Appendix 7, Tables A16–A33. The estimated coefficients are mostly the same indicating the robustness of the models' results.

Moreover, the FE estimator in simultaneous equations controlling for country fixed effect is also performed. The results are provided in Appendix 7. The estimated effect of health and education on growth is comparable to the estimates of 3SLS. The effect of physical capital per labor on growth has lower values than usual, although it is always significant and positive. The effects of both government size and trade openness are less significant in the case of FE than in the case of other estimates. As expected, there is a reduction in the estimated effect of democracy on the channel variables; as controlling for country effects is akin to ignoring some of the between-country variation in the determinants of the channels, which may drive much of their partial covariation with democracy. As a result, the effects of democracy on education, health, and government size are less significant than in other estimators. In addition, democracy has mostly a negative effect on investment. However, democracy has the same effect on trade openness as in the case of other estimators.

## 7. Conclusions, Policy Implications and Future Work

Our results conclude that the overall indirect effect of democracy on economic growth is significantly positive in the more democratic countries but turns negative in the less democratic countries within the MENA region. Therefore, the effect of democracy on economic growth is non-monotonic and changes according to the stage of democratic/political transition of countries. The effect of democracy on education and physical capital is responsible for this non-monotonic relationship. However, the effect of democracy on mortality rate and government size is always negative, and the effect of democracy on trade openness is always positive regardless the stage of democratic transition of different groups of countries.

Moreover, our results indicate that democracy might induce growth in rich countries, but hinders it in poor countries. A minimal level of income—$4000 per capita annually on average—was estimated as the lowest sufficient amount required for people to be educated and affluent enough to have the ability to both seek their political rights and to practice them effectively. Again, the effect of democracy on growth is found to be non-monotonic according to the standards of living in MENA countries. Therefore, improving economic circumstances, namely GDPP in MENA countries, as well as pursuing inclusive sustainable development policies need to be undertaken alongside adopting democratic governance since they are important fundamentals for positive democratic practice in these countries.

Human capital and physical capital accumulation were found to be the most effective links through which democracy affects growth in the MENA region. The health channel is more affected by the economic circumstances of the group of countries, whereas education is more affected by the democratic stage of the countries. Physical capital is highly affected by both the economic circumstances and political circumstances of the group of countries within the region.

It is recommended that policy makers encourage democracy in the relatively democratic groups of countries to stimulate economic growth. However, the autocratic groups of countries should start taking effective measures to transfer their regimes and their institutional framework towards one that can support real democratic practice, so they can reap the effects of democracy on growth.

Some areas in economic growth/democracy models have not been addressed in this study and need to be considered in future work. Electoral democracy is only one aspect of democracy, but institutional development ensures that the elected official is really empowered after election, they also guard against the possibility of elections being rigged. Democracy does not end at the ballot box. Therefore, it would be very beneficial to examine other measures or indices that capture other institutional aspects and components of democracy. In addition, the relationship between democracy and economic growth can be investigated in different groups of countries and for different time periods. Moreover, the indirect effect of democracy on economic growth can be estimated using other seemingly significant channels of influence, especially inflation, corruption and political instability as well as rule of law measures.

**Author Contributions:** Conceptualization, S.N.; Methodology, S.N.; Validation, S.N. and A.E.-K.; Formal Analysis, S.N.; Investigation, S.N. and A.E.-K.; Resources, S.N. and A.E.-K.; Data Curation, S.N. and A.E.-K.; Writing-Original Draft Preparation, S.N. and A.E.-K.; Writing-Review & Editing, S.N. and A.E.-K.; Visualization, S.N.; Supervision, S.N.; Project Administration, S.N.

**Funding:** This research received no external funding.

**Conflicts of Interest:** The authors declare no conflict of interest.

## Appendix A. Variables Specification

**Table A1.** Variables Definitions and Sources.

| Variable | Measurement | Data Source |
|---|---|---|
| Economic growth (GDPP) | Real per capita gross domestic product | World Bank (2016) |
| Health (MR) | Mortality rate, infant (per 1000 live births) | World Bank (2016) |
| Education (EDUS) | Education, School enrolment, secondary (% gross). | World Bank (2016) |
| Physical capital accumulation per labor (GCFL) | Gross capital formation includes land improvements; plant, machinery, and equipment purchases; and the construction of roads, railways, schools, offices, hospitals, commercial and industrial buildings divided by employment in each country | World Bank (2016) |
| Government size (GZ) | Government final consumption expenditure includes all government current expenditures for purchases of goods and services, expenditures on national defence and security (% of GDP) | World Bank (2016) |
| Total trade (TR) | Imports and exports of Egypt relative to GDP | World Bank (2016) |
| Electoral democracy (EDEM) | Achieved through electoral competition for the electorate's approval under circumstances when suffrage is extensive; political and civil society organizations can operate freely; elections are clean; and elections affect the composition of the chief executive of the country. In between elections, there is freedom of expression and an independent media capable of presenting alternative views on matters of political relevance. | Coppedge et al. (2016), V-DEM institute, University of Gothenburg, Sweden. |
| Primary Education (EDUP) | Education, School enrolment, primary (% gross). | World Bank (2016) |
| Female primary education (EDUPF) | School enrolment, primary, female (% gross) | World Bank (2016) |
| Exchange rate (EX) | Official exchange rate (LCU per US$, period average) | World Bank (2016) |
| Population growth (POPG) | Annual change in Population (%) | World Bank (2016) |
| Urban population (UPOP) | Urban population (% of total) | World Bank (2016) |
| Share of population (POP1) | Population ages 0–14 (% of total) | World Bank (2016) |
| Share of population (POP2) | Population ages 65 and above (% of total) | World Bank (2016) |
| Corruption Index (CO) | Corruption Index (V-DEM index) | Coppedge et al. (2016), V-DEM institute, University of Gothenburg, Sweden. |
| Unemployment (UM) | Unemployment, total (% of total labor force) | World Bank (2016) |
| Inflation (INF) | Inflation, consumer prices (annual %) | World Bank (2016) |
| Land area (LA) | Land area (sq. km) | World Bank (2016) |
| Oil producing countries (Dummy) | If the oil production is more than one million barrels per day, the dummy = 1, 0 otherwise | - |

## Appendix B. Models Specification and Descriptive Statistics

**Table A2.** Models Specification.

| Stages of Political Democracy | | | | | | Standards of Living | |
|---|---|---|---|---|---|---|---|
| Freedom House Classification | | V-DEM Classification | | Data Classification | | Average per Capita Income ($) | |
| Free and Partly Free | Not Free | Electoral and Liberal Democracy and Electoral Autocracy | Closed Autocracy | Democratic Countries | Autocratic Countries | Poor Countries | Rich Countries |
| Kuwait Lebanon Morocco Tunisia Turkey | Algeria Djibouti Egypt Iran Iraq Jordan Libya Oman Qatar Saudi Arabia Sudan Yemen | Algeria Djibouti Egypt Iran Iraq Lebanon Sudan Tunisia Turkey | Jordan Kuwait Libya Morocco Oman Qatar Saudi Arabia Yemen | Algeria Djibouti Iraq Kuwait Lebanon Tunisia Turkey Morocco Yemen | Egypt Iran Jordan Libya Oman Qatar Saudi Arabia Sudan | Algeria Djibouti Egypt Iraq Jordan Morocco Sudan Tunisia Yemen | Iran Kuwait Lebanon Libya Oman Qatar Saudi Arabia Turkey |

**Table A3.** Descriptive statistics in the variables in Free and partly free model (N = 5, T = 26), panel is balanced.

| Variable | Obs. | Mean | Std. Dev. | Min | Max |
|---|---|---|---|---|---|
| LNGDPP | 130 | 8.850 | 1.000 | 7.438 | 10.812 |
| LNMR | 130 | 3.014 | 0.596 | 1.960 | 4.145 |
| LNSES | 130 | 4.262 | 0.301 | 3.570 | 4.705 |
| LNGCM | 130 | 8.540 | 0.707 | 7.392 | 10.053 |
| LNGZ | 130 | 2.823 | 0.293 | 2.363 | 4.334 |
| LNTR | 130 | 4.283 | 0.348 | 3.417 | 4.960 |
| EDEM | 130 | 0.372 | 0.167 | 0.139 | 0.752 |

**Table A4.** Descriptive statistics in the variables in Not Free model (N = 12, T = 26), panel is balanced.

| Variable | Obs. | Mean | Std. Dev. | Min | Max |
|---|---|---|---|---|---|
| LNGDPP | 312 | 8.428 | 1.186 | 6.640 | 11.354 |
| LNMR | 312 | 3.322 | 0.644 | 1.917 | 4.529 |
| LNSES | 312 | 4.135 | 0.540 | 2.262 | 5.087 |
| LNGCM | 312 | 8.139 | 1.215 | 4.042 | 10.862 |
| LNGZ | 312 | 2.809 | 0.475 | 0.847 | 4.041 |
| LNTR | 312 | 4.108 | 0.963 | −3.863 | 5.430 |
| EDEM | 312 | 0.201 | 0.118 | 0.016 | 0.526 |

**Table A5.** Descriptive statistics in the variables in Electoral and liberal democracy and electoral autocracy model (N = 9, T = 26), panel is balanced.

| Variable | Obs. | Mean | Std. Dev. | Min | Max |
|---|---|---|---|---|---|
| LNGDPP | 234 | 8.082 | 0.701 | 6.640 | 9.539 |
| LNMR | 234 | 3.426 | 0.588 | 1.960 | 4.529 |
| LNSES | 234 | 4.037 | 0.517 | 2.262 | 4.706 |
| LNGCM | 234 | 7.842 | 0.860 | 5.741 | 9.370 |
| LNGZ | 234 | 2.635 | 0.428 | 0.847 | 3.746 |
| LNTR | 234 | 3.951 | 1.069 | −3.863 | 5.430 |
| EDEM | 234 | 0.314 | 0.159 | 0.087 | 0.752 |

**Table A6.** Descriptive statistics in the variables in closed autocracy model (N = 8, T = 26), panel is balanced.

| Variable | Obs. | Mean | Std. Dev. | Min | Max |
|---|---|---|---|---|---|
| LNGDPP | 208 | 9.080 | 1.316 | 6.649 | 11.354 |
| LNMR | 208 | 3.013 | 0.639 | 1.917 | 4.485 |
| LNSES | 208 | 4.325 | 0.397 | 3.115 | 5.087 |
| LNGCM | 208 | 8.723 | 1.165 | 4.042 | 10.862 |
| LNGZ | 208 | 3.013 | 0.331 | 2.320 | 4.334 |
| LNTR | 208 | 4.394 | 0.303 | 3.380 | 5.007 |
| EDEM | 208 | 0.180 | 0.114 | 0.016 | 0.526 |

**Table A7.** Descriptive statistics in the variables in Democratic-countries model (N = 9, T = 26), panel is balanced.

| Variable | Obs. | Mean | Std. Dev. | Min | Max |
|---|---|---|---|---|---|
| LNGDPP | 234 | 8.315 | 1.039 | 6.649 | 10.812 |
| LNMR | 234 | 3.374 | 0.664 | 1.960 | 4.529 |
| LNSES | 234 | 4.015 | 0.529 | 2.262 | 4.706 |
| LNGCM | 234 | 7.923 | 1.028 | 4.042 | 10.053 |
| LNGZ | 234 | 2.822 | 0.391 | 0.847 | 4.334 |
| LNTR | 234 | 4.167 | 1.033 | −3.863 | 5.430 |
| EDEM | 234 | 0.340 | 0.144 | 0.087 | 0.752 |

**Table A8.** Descriptive statistics in the variables in Autocratic-countries model (N = 8, T = 26), panel is balanced.

| Variable | Obs. | Mean | Std. Dev. | Min | Max |
|---|---|---|---|---|---|
| LNGDPP | 208 | 8.818 | 1.210 | 6.640 | 11.354 |
| LNMR | 208 | 3.071 | 0.585 | 1.917 | 4.381 |
| LNSES | 208 | 4.349 | 0.357 | 3.507 | 5.087 |
| LNGCM | 208 | 8.632 | 1.070 | 6.093 | 10.862 |
| LNGZ | 208 | 2.803 | 0.469 | 1.522 | 4.041 |
| LNTR | 208 | 4.152 | 0.532 | 2.406 | 5.007 |
| EDEM | 208 | 0.151 | 0.092 | 0.016 | 0.526 |

## Appendix C. Diagnostic Tests

**Table A9.** Hausman test of Endogeneity.

| Effect of Channel on Growth | $\chi^2_1$ | *p*-Value |
|---|---|---|
| Free and partly free | 86.11 | 0.00 |
| Not free | 24.95 | 0.00 |
| Electoral and liberal democracy and electoral autocracy | 19.51 | 0.00 |
| Closed autocracy | 31.86 | 0.00 |
| Democratic-countries | 57.45 | 0.00 |
| Autocratic-countries | 45.82 | 0.00 |

The null hypothesis H0: difference in coefficients between OLS and 3SLS are not systematic.

**Table A10.** Multicollinearity tests (mean VIF).

| | EQ (1) | EQ (2) | EQ (3) | EQ (4) | EQ (5) | EQ (6) |
|---|---|---|---|---|---|---|
| Free and partly free | 6.21 | 1.78 | 1.09 | 3.72 | 9.82 | 19.31 |
| Not free | 4.10 | 4.97 | 1.79 | 5.52 | 2.33 | 2.52 |
| Electoral and liberal democracy and electoral autocracy | 8.88 | 7.06 | 6.57 | 1.59 | 3.79 | 2.03 |
| Closed autocracy | 5.016 | 11.09 | 1.48 | 2.07 | 1.57 | 3.88 |
| Democratic-countries | 3.35 | 5.13 | 1.71 | 2.04 | 1.61 | 2.48 |
| Autocratic-countries | 4.82 | 1.82 | 2.47 | 1.36 | 3.35 | 5.15 |

## Appendix D. Robustness Analysis

**Table A11.** The Effect of the Channels on Economic Growth in Different Models (1995–2015).

| Effect of Channel on Growth | Education | Health | Ph. Capital | Gov. Size | Trade |
|---|---|---|---|---|---|
| **Free and partly free** | 0.442 [3.32] | −0.296 [−2.98] | 1.165 [21.14] | 0.878 [5.47] | −0.316 [−3.07] |
| **Not free** | 0.298 [4.27] | −0.989 [−16.27] | 0.824 [25.35] | 0.520 [6.34] | −0.223 [−5.98] |
| **Electoral and liberal democracy and electoral autocracy** | 0.569 [8.94] | −0.687 [−13.22] | 0.494 [13.77] | 0.495 [5.49] | −0.205 [−6.08] |
| **Closed autocracy** | 0.385 [3.31] | −1.255 [−15.20] | 1.067 [27.27] | 0.253 [2.28] | −0.416 [−3.24] |
| **Democratic-countries** | 0.555 [6.63] | −0.615 [−8.98] | 0.705 [17.25] | 0.604 [5.56] | −0.253 [−6.89] |
| **Autocratic-countries** | 0.386 [3.33] | −0.815 [−9.44] | 0.952 [26.01] | 0.546 [5.98] | −0.266 [−2.94] |

T-statistics are included in Parentheses.

**Table A12.** The Effect of Democracy on the Channels in Different Models (1995–2015).

| Effect of Democracy on the Channel | Education | Health | Ph. Capital | Gov. Size | Trade |
|---|---|---|---|---|---|
| **Free and partly free** | 0.265 [3.79] | −1.121 [−3.73] | 0.480 [3.72] | −0.622 [−3.06] | 0.879 [5.00] |
| **Not free** | −0.634 [−5.48] | −0.127 [−0.79] | −1.040 [−3.27] | 0.745 [3.07] | 1.444 [2.75] |
| **Electoral and liberal democracy and electoral autocracy** | 0.125 [1.08] | −0.789 [−4.65] | 1.190 [4.26] | −0.770 [−2.81] | 2.009 [3.68] |
| **Closed autocracy** | −0.722 [−5.34] | −0.849 [−5.52] | −0.937 [−2.77] | −0.282 [−1.46] | 0.393 [1.84] |
| **Democratic-countries** | 0.230 [2.58] | −0.957 [−3.96] | 0.525 [2.10] | −0.579 [−4.37] | 1.607 [3.31] |
| **Autocratic-countries** | −0.483 [−4.93] | −2.246 [−9.56] | −1.854 [−4.44] | −1.204 [−3.75] | 0.490 [1.10] |

T-statistics are included in Parentheses.

**Table A13.** The Effect of Democracy on the Channels in Different Models—lnGDP.

| Effect of Democracy on the Channel | Education | Health | Ph. Capital | Gov. Size | Trade |
|---|---|---|---|---|---|
| **Free and partly free** | 0.209 [3.06] | −0.989 [−3.03] | 0.515 [3.80] | −0.483 [−2.13] | 0.455 [2.73] |
| **Not free** | −0.623 [−5.50] | −0.186 [−1.21] | −1.183 [−3.77] | 0.774 [3.32] | 1.986 [3.55] |
| **Electoral and liberal democracy and electoral autocracy** | 0.039 [0.36] | −0.827 [−5.14] | 1.216 [4.81] | −0.723 [−2.87] | 2.512 [4.28] |
| **Closed autocracy** | −0.879 [−6.46] | −0.916 [−6.00] | −1.279 [−3.91] | −0.435 [−2.34] | 0.340 [1.61] |
| **Democratic-countries** | 0.205 [2.34] | −0.771 [−3.23] | 0.708 [2.86] | −0.536 [−4.22] | 2.691 [5.18] |
| **Autocratic-countries** | −0.422 [−4.26] | −1.984 [−8.42] | −2.031 [−4.86] | −1.192 [−3.85] | 0.801 [1.79] |

T-statistics are included in Parentheses.

**Table A14.** The Effect of Democracy on the Channels in Different Models—lnGDPG.

| Effect of Democracy on the Channel | Education | Health | Ph. Capital | Gov. Size | Trade |
|---|---|---|---|---|---|
| **Free and partly free** | 0.255 [3.38] | −1.097 [−3.24] | 0.606 [4.34] | −0.564 [−2.15] | 0.549 [3.01] |
| **Not free** | −0.510 [−4.49] | 0.080 [0.49] | −1.317 [−4.30] | 0.796 [3.24] | 1.972 [3.00] |
| **Electoral and liberal democracy and electoral autocracy** | −0.076 [−0.67] | −0.910 [−5.37] | 1.660 [5.83] | −0.414 [−1.65] | 3.194 [4.79] |
| **Closed autocracy** | −0.826 [−5.95] | −0.818 [−4.76] | −1.646 [−6.00] | −0.066 [−0.33] | 0.100 [0.45] |
| **Democratic-countries** | 0.225 [2.52] | −0.815 [−3.14] | 0.808 [3.40] | −0.623 [−4.44] | 2.896 [5.08] |
| **Autocratic-countries** | −0.263 [−2.6] | −1.790 [−6.69] | −2.330 [−5.47] | −0.940 [−3.01] | 0.934 [1.91] |

T-statistics are included in Parentheses.

**Table A15.** The Effect of Democracy on the Channels in Different Models—lnGDPPG.

| Effect of Democracy on the Channel | Education | Health | Ph. Capital | Gov. Size | Trade |
|---|---|---|---|---|---|
| **Free and partly free** | 0.256 [3.01] | −0.669 [−1.72] | 0.497 [3.25] | −0.806 [−2.37] | 0.241 [1.38] |
| **Not free** | −0.427 [−3.56] | 0.397 [2.61] | −1.187 [−3.68] | 0.778 [2.86] | 1.612 [2.36] |
| **Electoral and liberal democracy and electoral autocracy** | 0.117 [0.95] | −0.391 [−2.43] | 1.561 [5.38] | −0.769 [−2.44] | 2.677 [4.08] |
| **Closed autocracy** | −0.766 [−4.84] | −0.559 [−3.41] | −1.385 [−4.38] | −0.151 [−0.65] | −0.045 [−0.18] |
| **Democratic-countries** | 0.316 [3.59] | −0.491 [−1.88] | 0.786 [3.19] | −0.364 [−2.27] | 2.310 [3.76] |
| **Autocratic-countries** | −0.230 [−7.71] | −0.965 [−3.46] | −2.206 [−4.35] | −0.844 [−2.26] | 0.884 [1.61] |

T-statistics are included in Parentheses.

**Table A16.** The Effect of the Channel on Economic Growth in Free and partly free using different methods of estimation.

| Effect of Channel on Growth | Education | Health | Ph. Capital | Gov. Size | Trade | $R^2$ |
|---|---|---|---|---|---|---|
| **3SLS** | 0.575 [4.53] | −0.301 [−3.32] | 1.111 [20.52] | 0.893 [5.82] | −0.282 [−2.94] | 0.89 |
| **2SLS** | 0.324 [2.1] | −0.248 [−2.24] | 1.211 [19.35] | 0.563 [3.1] | −0.020 [−0.18] | 0.90 |
| **SUR** | 0.429 [3.32] | −0.249 [−2.66] | 1.161 [21.32] | 0.657 [4.97] | −0.171 [1.84] | 0.90 |
| **OLS** | 0.243 [1.62] | −0.206 [−1.91] | 1.240 [20.35] | 0.280 [1.83] | 0.075 [0.70] | 0.91 |
| **FE** | 0.501 [10.82] | −0.349 [−11.63] | 0.205 [4.99] | −0.095 [−1.68] | 0.223 [4.61] | 0.74 |

T-statistics are included in Parentheses.

**Table A17.** The Effect of Democracy on the Channels in Free and partly free model.

| Effect of Democracy on the Channel | Education | Health | Ph. Capital | Gov. Size | Trade |
|---|---|---|---|---|---|
| **3SLS** | 0.239 [3.33] | −1.029 [−3.14] | 0.499 [3.75] | −0.573 [−2.89] | 1.003 [5.16] |
| **2SLS** | 0.227 [3.21] | −0.865 [−2.57] | 0.590 [4.20] | −1.025 [−3.6] | 0.746 [3.44] |
| **SUR** | 0.233 [3.37] | −1.022 [−3.16] | 0.450 [3.44] | −0.534 [−2.52] | 0.655 [3.72] |
| **OLS** | 0.227 [3.21] | −0.864 [−2.58] | 0.555 [3.97] | −0.712 [−2.64] | 0.457 [2.38] |
| **FE** | −0.077 [−0.90] | −0.263 [−2.38] | −0.374 [−2.77] | −1.397 [−3.38] | 0.429 [3.11] |

T-statistics are included in.

**Table A18.** The Indirect Effect of Democracy on Economic Growth in Free and partly free model.

| DEM/EG (%) | Total | Education | Health | Ph. Capital | Gov. Size | Trade |
|---|---|---|---|---|---|---|
| **3SLS** | 0.207 | 0.138 [2.75] | 0.310 [2.28] | 0.554 [3.60] | −0.512 [−2.77] | −0.283 [−0.93] |
| **2SLS** | 0.410 | 0.074 [1.75] | 0.214 [2.00] | 0.715 [4.11] | −0.578 [−2.35] | −0.015 [−0.18] |
| **SUR** | 0.415 | 0.100 [2.37] | 0.255 [2.03] | 0.523 [3.40] | −0.351 [−2.25] | −0.112 [−1.82] |
| **OLS** | 0.755 | 0.055 [1.45] | 0.178 [1.53] | 0.688 [3.90] | −0.200 [−1.50] | 0.034 [0.67] |

T-statistics of the indirect effect coefficients are calculated using the delta method.

**Table A19.** The Effect of the Channel on Economic Growth in Not Free model using different methods of estimation.

| Effect of Channel on Growth | Education | Health | Ph. Capital | Gov. Size | Trade | $R^2$ |
|---|---|---|---|---|---|---|
| **3SLS** | 0.269 [4.19] | −0.935 [−16.44] | 0.833 [27.19] | 0.505 [6.76] | −0.224 [−6.47] | 0.86 |
| **2SLS** | 0.216 [3.20] | −0.698 [−10.72] | 0.848 [26.22] | 0.488 [6.02] | −0.258 [−6.97] | 0.86 |
| **SUR** | 0.278 [4.42] | −0.911 [−15.80] | 0.828 [27.08] | 0.504 [6.77] | −0.223 [−6.50] | 0.86 |
| **OLS** | 0.216 [3.20] | −0.698 [−10.72] | 0.848 [26.22] | 0.488 [6.02] | −0.258 [−6.97] | 0.86 |
| **FE** | 0.202 [3.95] | −0.295 [−7.32] | 0.159 [5.63] | −0.167 [−4.05] | 0.044 [2.55] | 0.33 |

T-statistics are included in Parentheses.

**Table A20.** The Effect of Democracy on the Channels in Not Free model.

| Effect of Democracy on the Channel | Education | Health | Ph. Capital | Gov. Size | Trade |
|---|---|---|---|---|---|
| **3SLS** | −0.669 [−5.82] | −0.317 [−2.02] | −0.971 [−3.17] | 0.969 [3.97] | 2.199 [3.89] |
| **2SLS** | −0.702 [−5.92] | −0.320 [2.01] | −0.913 [−2.78] | 0.723 [2.89] | 2.223 [3.82] |
| **SUR** | −0.550 [−4.95] | −0.190 [−1.23] | −0.701 [−2.48] | 0.728 [3.05] | 2.049 [3.67] |
| **OLS** | −0.641 [−5.56] | −0.183 [−1.17] | −0.861 [−2.84] | 0.482 [1.97] | 2.082 [3.64] |
| **FE** | 0.206 [1.92] | −0.460 [−2.86] | −0.195 [−0.63] | 1.878 [7.03] | 3.657 [5.40] |

T-statistics are included in Parentheses.

**Table A21.** The Indirect Effect of Democracy on Economic Growth in Not Free model.

| DEM/EG (%) | Total | Education | Health | Ph. Capital | Gov. Size | Trade |
|---|---|---|---|---|---|---|
| **3SLS** | −0.693 | −0.177 [−3.40] | 0.296 [2.00] | −0.790 [−3.15] | 0.489 [3.42] | −0.493 [−3.33] |
| **2SLS** | −0.924 | −0.152 [−2.81] | 0.223 [1.97] | −0.774 [−2.77] | 0.353 [2.60] | −0.574 [−3.35] |
| **SUR** | −0.651 | −0.153 [−3.29] | 0.173 [1.22] | −0.580 [−2.47] | 0.367 [2.78] | −0.458 [−3.20] |
| **OLS** | −1.044 | −0.139 [−2.77] | 0.128 [1.17] | −0.730 [−2.82] | 0.235 [1.88] | −0.538 [−3.23] |

T-statistics of the indirect effect coefficients are calculated using the delta method.

**Table A22.** The Effect of the Channel on Economic Growth in Electoral and liberal Democracy and electoral autocracy model using different methods of estimation.

| Effect of Channel on Growth | Education | Health | Ph. Capital | Gov. Size | Trade | $R^2$ |
|---|---|---|---|---|---|---|
| **3SLS** | 0.550 [9.39] | −0.689 [−14.14] | 0.514 [15.21] | 0.531 [6.25] | −0.215 [−6.75] | 0.79 |
| **2SLS** | 0.526 [8.80] | −0.501 [−9.61] | 0.542 [15.7] | 0.546 [6.26] | −0.217 [−6.61] | 0.79 |
| **SUR** | 0.601 [10.37] | −0.687 [−14.10] | 0.491 [14.69] | 0.524 [6.29] | −0.197 [−6.29] | 0.79 |
| **OLS** | 0.526 [8.80] | −0.501 [−9.61] | 0.542 [15.70] | 0.546 [6.62] | −0.217 [−6.61] | 0.79 |
| **FE** | 0.380 [7.54] | −0.449 [−15.76] | 0.148 [4.63] | 0.086 [1.54] | −0.011 [−0.64] | 0.43 |

T-statistics are included in Parentheses.

**Table A23.** The Effect of Democracy on the Channels in Electoral and liberal Democracy and electoral autocracy model.

| Effect of Democracy on the Channel | Education | Health | Ph. Capital | Gov. Size | Trade |
|---|---|---|---|---|---|
| **3SLS** | 0.166 [1.49] | −0.830 [−5.10] | 1.258 [4.65] | −0.779 [−3.02] | 3.019 [4.84] |
| **2SLS** | 0.338 [2.69] | −0.883 [−5.37] | 1.463 [5.29] | −0.276 [−0.90] | 2.773 [4.12] |
| **SUR** | 0.193 [1.89] | −0.869 [−5.34] | 1.177 [4.36] | 0.337 [1.62] | 2.761 [4.55] |
| **OLS** | 0.285 [2.64] | −0.883 [−5.37] | 1.463 [5.29] | 0.404 [1.89] | 2.102 [3.37] |
| **FE** | 0.221 [2.50] | 0.027 [0.18] | −0.526 [−2.20] | 0.071 [0.33] | 1.067 [1.69] |

T-statistics are included in Parentheses.

**Table A24.** The Indirect Effect of Democracy on Economic Growth in Electoral and liberal Democracy and electoral autocracy model.

| DEM/EG (%) | Total | Education | Health | Ph. Capital | Gov. Size | Trade |
|---|---|---|---|---|---|---|
| **3SLS** | 0.244 | 0.089 [1.36] | 0.572 [4.80] | 0.647 [4.45] | −0.413 [−2.72] | −0.650 [−3.94] |
| **2SLS** | 0.661 | 0.178 [2.57] | 0.442 [4.69] | 0.793 [5.01] | −0.151 [−0.89] | −0.601 [−3.49] |
| **SUR** | 0.923 | 0.116 [1.86] | 0.597 [4.99] | 0.577 [4.18] | 0.176 [1.57] | −0.544 [−3.69] |
| **OLS** | 1.150 | 0.150 [2.53] | 0.442 [4.69] | 0.793 [5.01] | 0.220 [1.81] | −0.456 [−3.00] |

T-statistics of the indirect effect coefficients are calculated using the delta method.

**Table A25.** The Effect of the Channel on Economic Growth in Closed Autocracy model using different methods of estimation.

| Effect of Channel on Growth | Education | Health | Ph. Capital | Gov. Size | Trade | $R^2$ |
|---|---|---|---|---|---|---|
| **3SLS** | 0.285 [2.71] | −1.179 [−15.63] | 1.086 [29.00] | 0.209 [2.03] | −0.427 [−3.70] | 0.90 |
| **2SLS** | 0.186 [1.68] | −0.952 [11.14] | 1.120 [28.91] | 0.150 [1.38] | −0.549 [−4.53] | 0.90 |
| **SUR** | 0.291 [2.76] | −1.168 [15.3] | 1.087 [29.01] | 0.195 [1.89] | −0.425 [−3.67] | 0.90 |
| **OLS** | 0.186 [1.68] | −0.952 [−11.14] | 1.120 [28.91] | 0.150 [1.38] | −0.549 [−4.53] | 0.90 |
| **FE** | 0.159 [2.34] | −0.126 [−2.29] | 0.159 [4.23] | −0.229 [−5.25] | 0.046 [0.485] | 0.38 |

T-statistics are included in Parentheses.

**Table A26.** The Effect of Democracy on the Channels in Closed Autocracy model.

| Effect of Democracy on the Channel | Education | Health | Ph. Capital | Gov. Size | Trade |
|---|---|---|---|---|---|
| **3SLS** | −0.752 [−5.51] | −1.041 [−6.69] | −0.865 [−2.67] | −0.383 [−2.07] | 0.557 [2.61] |
| **2SLS** | −0.817 [−5.73] | −0.960 [6.07] | −1.044 [−3.08] | −0.446 [−2.34] | 0.632 [2.66] |
| **SUR** | −0.612 [−4.70] | −1.025 [−6.63] | −0.897 [−2.80] | −0.308 [−1.72] | 0.554 [2.63] |
| **OLS** | −0.756 [−5.55] | −0.936 [−5.96] | −1.188 [−3.56] | −0.370 [−2.00] | 0.590 [2.54] |
| **FE** | −0.300 [−1.82] | −1.880 [−7.03] | 0.141 [0.30] | 1.530 [4.39] | 0.580 [2.19] |

T-statistics are included in Parentheses.

**Table A27.** The Indirect Effect of Democracy on Economic Growth in Closed Autocracy model using different methods of estimation.

| DEM/EG (%) | Total | Education | Health | Ph. Capital | Gov. Size | Trade |
|---|---|---|---|---|---|---|
| **3SLS** | −0.244 | −0.215 [−2.43] | 1.227 [6.15] | −0.939 [−2.66] | −0.080 [−1.33] | −0.237 [−12.63] |
| **2SLS** | −0.822 | −0.152 [−1.61] | 0.914 [−3.06] | −1.170 [−3.06] | −0.067 [−1.19] | −0.347 [−2.30] |
| **SUR** | −0.251 | −0.178 [−2.38] | 1.197 [6.08] | −0.975 [−2.79] | −0.060 [−1.27] | −0.235 [−2.14] |
| **OLS** | −0.961 | −0.141 [−1.61] | 0.891 [5.25] | −1.331 [−3.53] | −0.055 [−1.13] | −0.324 [−2.21] |

T-statistics of the indirect effect coefficients are calculated using the delta method.

**Table A28.** The Effect of the Channel on Economic Growth in Democratic-countries model using different methods of estimation.

| Effect of Channel on Growth | Education | Health | Ph. Capital | Gov. Size | Trade | $R^2$ |
|---|---|---|---|---|---|---|
| **3SLS** | 0.525 [6.99] | −0.599 [−9.54] | 0.723 [18.93] | 0.595 [5.90] | −0.253 [−7.39] | 0.85 |
| **2SLS** | 0.580 [6.86] | −0.682 [−10.28] | 0.770 [17.88] | 0.755 [6.59] | −0.312 [−7.96] | 0.85 |
| **SUR** | 0.592 [7.94] | −0.597 [9.44] | 0.695 [18.30] | 0.638 [6.37] | −0.252 [−7.41] | 0.85 |
| **OLS** | 0.580 [6.86] | −0.682 [−10.28] | 0.770 [17.88] | 0.755 [6.59] | −0.312 [−7.96] | 0.85 |
| **FE** | 0.363 [9.06] | −0.391 [−12.88] | 0.110 [4.08] | 0.034 [0.79] | 0.001 [0.07] | 0.46 |

T-statistics are included in Parentheses.

**Table A29.** The Effect of Democracy on the Channels in Democratic-countries model.

| Effect of Democracy on the Channel | Education | Health | Ph. Capital | Gov. Size | Trade |
|---|---|---|---|---|---|
| **3SLS** | 0.250 [2.91] | −0.868 [−3.73] | 0.636 [2.71] | −0.585 [−4.43] | 2.543 [4.86] |
| **2SLS** | 0.280 [3.20] | −0.810 [−3.25] | 0.673 [2.66] | −0.529 [−3.91] | 2.139 [3.95] |
| **SUR** | 0.320 [3.87] | −0.868 [−3.73] | 0.654 [2.80] | −0.558 [−4.38] | 2.629 [5.06] |
| **OLS** | 0.313 [3.70] | −0.810 [−3.25] | 0.741 [2.94] | −0.500 [−3.84] | 2.253 [4.20] |
| **FE** | 0.099 [0.95] | −0.183 [−1.28] | −0.721 [−3.02] | 0.258 [1.48] | 0.046 [0.22] |

T-statistics are included in Parentheses.

**Table A30.** The Indirect Effect of Democracy on Economic Growth in MENA Countries in Democratic-countries model.

| DEM/EG (%) | Total | Education | Health | Ph. Capital | Gov. Size | Trade |
|---|---|---|---|---|---|---|
| **3SLS** | 0.120 | 0.131 [2.69] | 0.521 [1.00] | 0.460 [2.68] | −0.348 [−3.54] | −0.644 [−10.81] |
| **2SLS** | 0.167 | 0.162 [2.90] | 0.553 [3.10] | 0.518 [2.63] | −0.400 [−3.36] | −0.667 [−3.54] |
| **SUR** | 0.145 | 0.190 [3.48] | 0.518 [10.27] | 0.455 [2.77] | −0.356 [−3.61] | −0.662 [−4.18] |
| **OLS** | 0.225 | 0.182 [3.26] | 0.553 [3.10] | 0.571 [2.90] | −0.378 [−3.32] | −0.703 [−3.71] |

T-statistics of the indirect effect coefficients are calculated using the delta method.

**Table A31.** The Effect of the Channel on Economic Growth in Autocratic-countries model using different methods of estimation.

| Effect of Channel on Growth | Education | Health | Ph. Capital | Gov. Size | Trade | R$^2$ |
|---|---|---|---|---|---|---|
| **3SLS** | 0.272 [2.57] | −0.762 [−9.62] | 0.955 [27.11] | 0.546 [6.55] | −0.246 [−3.01] | 0.88 |
| **2SLS** | 0.244 [2.10] | −0.755 [−9.11] | 1.026 [27.42] | 0.344 [3.64] | −0.297 [−3.24] | 0.89 |
| **SUR** | 0.291 [2.76] | −0.755 [−9.53] | 0.946 [27.01] | 0.541 [6.50] | −0.241 [−2.95] | 0.88 |
| **OLS** | 0.244 [2.10] | −0.755 [−9.11] | 1.026 [27.42] | 0.344 [3.64] | −0.297 [−3.24] | 0.89 |
| **FE** | 0.255 [2.77] | −0.241 [−4.92] | 0.181 [4.03] | −0.171 [−3.5] | 0.203 [3.24] | 0.44 |

T-statistics are included in Parentheses.

**Table A32.** The Effect of Democracy on the Channels in Autocratic-countries model using different methods of estimation.

| Effect of Democracy on the Channel | Education | Health | Ph. Capital | Gov. Size | Trade |
|---|---|---|---|---|---|
| **3SLS** | −0.533 [−5.42] | −2.448 [−9.98] | −1.787 [−4.64] | −1.454 [−4.62] | 1.126 [2.60] |
| **2SLS** | −0.511 [−5.05] | −1.961 [−7.54] | −2.020 [−5.04] | −1.316 [−4.02] | 1.494 [3.31] |
| **SUR** | −0.546 [−5.63] | −2.220 [−9.40] | −1.711 [−4.44] | −1.197 [−4.05] | 1.074 [2.56] |
| **OLS** | −0.535 [−5.37] | −1.883 [−7.49] | −2.020 [−5.04] | −1.279 [−4.17] | 1.221 [2.79] |
| **FE** | −0.010 [0.06] | −0.648 [−2.29] | −0.131 [−0.32] | 1.126 [2.73] | 1.234 [4.37] |

T-statistics are included in Parentheses.

**Table A33.** The Indirect Effect of Democracy on Economic Growth in MENA Countries in Autocratic-countries model.

| DEM/EG (%) | Total | Education | Health | Ph. Capital | Gov. Size | Trade |
|---|---|---|---|---|---|---|
| **3SLS** | −1.413 | −0.145 [−2.32] | 1.864 [6.93] | −1.707 [−4.57] | −0.793 [−3.13] | −0.632 [−2.53] |
| **2SLS** | −1.611 | −0.125 [−1.94] | 1.481 [5.81] | −2.072 [−4.96] | −0.453 [−2.70] | −0.443 [−2.31] |
| **SUR** | −1.008 | −0.159 [−2.48] | 1.675 [6.70] | −1.618 [−4.39] | −0.647 [−3.44] | −0.259 [−1.94] |
| **OLS** | −1.582 | −0.130 [−1.96] | 1.422 [5.79] | −2.072 [−4.96] | −0.440 [−2.74] | −0.362 [−2.12] |

T-statistics are included in Parentheses.

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
