# Peer review of "The Indirect Effect of Democracy on Economic Growth in the MENA Region (1990–2015)"

_economies, doi:10.3390/economies6040061_

Reviewer 1 Report

Reviewer Report

The Indirect Effect of Democracy on Economic Growth in the MENA Region (1990-2015)

Economies Journal MDPI

 Summary:

The paper examines the effects of indirect channels of democracy on economic growth using a recent dataset for the MENA region. The authors use three stage least squares (3-SLS) as a main model for estimation and inference and conduct various robustness tests, including tests that justify the use of the model. The paper concludes that education and physical assets are responsible for a non-monotonic relationship between democracy and economic growth. Unlike trade openness which is impacted positively by democracy, the mortality rate and government size are impacted negatively.  Human and physical capital facilitate growth. The latter is affected by political and economic circumstances. The paper identifies the most effective channels through which democracy impacts growth.

Main Comments:

1-      I expect equation 7 to contain a covariance component which may not be trivial. The author(s) did not provide an explanation for the equation provided or show a derivation for the form used in the paper. Oehlert (1992) provided a method that could be used in here but his original paper, Oelhert did not derive a form that is applied to the model presented by the author(s). Hence, a justification is necessary.

2-      Log transformation does not imply normality.

3-      The authors have tested for endogeneity to justify the use of 3SLS which is appropriate in this case. A brief discussion; however, of identification and the way instruments are specified may further clarify the appropriateness of using 3SLS relative to other methods.

4-      For some models, the assessment of the goodness of fit as significant and strong is not accurate. Table 1 contains results of a model where the R-squared is zero.  Essentially, how R-squared is calculated is not clear. Was this done using actual values or endogenous instruments.

5-      For clarity, it is best to refer to the classification of democracy as indicated in the first column of Tables when discussing the results (See Table 1 for example against the analysis of results). When using the terms “more democratic” and “less democratic” on lines 300 and 301, what do the author(s) mean exactly? Are the author(s) referring to electoral and liberal democracies or democratic countries?

6-      The author(s) plot Figure 1 and discuss non-monotonicity in the MENA region, and then use the score of V-DEM, which is constructed for a larger set of countries, to compare with. It is not clear whether the score of 0.25 is for the MENA region and whether this score is calculated for the same sample period of the study.

7-      What is Total in Table 3 and where is the corresponding equation given? The notes state that “T-statistics of the total effect is the average of t-states of the different channels” and if so, this is not sound practice.

8-      Political stability is one of the variables that was mentioned by the author(s) in the conclusion and impacts both democracy and economic growth especially during the estimation period. I suggest controlling for political stability is the system of equations.

9-      The author(s) did not comment on potential changes of countries from one classification of democracy to another during the estimation period and the potential impact on results. Instability and conflict are likely to change democracy classification of countries over time; for example, in the MENA region during the sample period where the Arab spring and other conflicts occurred.

10-  Results should be discussed consistently in the abstract, the analysis and conclusion.

 11-  Last, I recommend that the author(s) control for country fixed effects, time fixed effect and linear and quadratic trends and/or show robustness using another estimation method.

 Other comments:

·         It is customary to report a summary of the main findings in the introduction.

·         Some coefficients discussed in the text were rounded while others were not.

·         The article needs proof reading. Some typos are detected in the text. For example, “no” on line 69; “using the delta methods” and “average t-states” in the notes to Table 3 and many other errors…

·         Some grammatical errors could also be avoided.

·         It is preferred to make a reference for each paper separately in the case of (Acemoglu, et al., 2001, 2002). The same author(s) had 2 different papers.

·         The reference list should be checked for errors. For example, I did not find Zghidi (2017) in the reference list.

Author Response

Dear Professor,

Thank you very much for your insightful comments, we have strived to perform all the required adjustments as best as we could. Kindly find attached a word file with point by point discussion of the performed revisions.

Reviewer 2 Report

Referee report

The Indirect Effect of Democracy on Economic Growth in the MENA Region (1990-2015) 

In this paper, the authors studythe impact of democracy on economic growth between 1990 to 2015. Using six simultaneous equations, 3SLS, the results indicate that democracy enhances growth through its positive effect on health in all classifications of countries within the MENA region. However, the effect of democracy on growth through education and physical capital/labor is non-monotonic. Besides the effect of democracy on growth is negative in less democratic countries and poor countries, but positive in more democratic countries and rich countries. 

The authors conduct a solid empirical investigation and are clearly masters of the sub-literature in this area. They employed a valid technique to the validity of the impact of democracy on economic growth. They did a good job at all motivating the theoretical apparatus they used. Since the results of this paper are very like the previous research.

That means there are no further contributions to the academic field or real business. The length of the paper is too long. The author(s) should rewrite the manuscript concisely. I recommend not to quote the results of other previous researches in empirical results and to reduce section 2.1 and 2.2. Major findings and the differences between this paper and the others are enough to describe in section 3. Emphasize to describe the differences and unique contributions compared to the previous research.The authors should highlight the importance of their results not only by enumerating the comparison with lasts studies but also by adding robust explanations to their findings. What do we conclude from the literature review with respect to the open questions and how will using the  3SLSmethod provide substantial new insight?  

 Author Response

Dear Professor,

Thank you very much for your insightful comments. We have strived to perform the required adjustments as best as we could. Kindly find attached a point-by-point reply to the required revisions.

Round  2

Reviewer 1 Report

Reviewer Report 2

The Indirect Effect of Democracy on Economic Growth in the MENA Region (1990-2015)

Economies Journal MDPI

  The updated version of the paper is improved relative to the first version. There are several comments provided in the earlier reviewer report that have been addressed. There are also changes in the new version that require new comments.

Below you find comments on modelling, interpretation and the presentation of the paper.

1-     I suggest that the authors would provide a summary of the findings in the introduction as it is the case in any scientific journal.

2-     The model on line 208 is well used in the literature. It is recommended to provide a reference of your choice in terms of the model structure. Some papers use an analytical exposition of the model others use an econometric specification like yourself. Since this is not a proposed model, the authors may state a paper they follow.

3-     With respect to using the delta method following Oehlert (1992), I suggest that the authors use the argument discussed by Baum and Lake (2003) in their article that you referred to in your response to my previous comments. The argument is on p.341 and it addresses concerns which I have raised in the previous report including the use of a FE model (with time effects as well).

4-     Line 344: The authors should clarify in a foot note how variables with VIF factors above 10 have been dealt with.

5-     The authors must explicitly note how standard errors of estimates were treated under each model. This could be added in the notes to tables, in the text or footnotes.

6-     The authors report the number of observations in each equation (channel).

7-     The authors should report how elasticities are calculated via each channel and provide the equation used in the analysis of Table 1. It is also important to differentiate between the direct effect and channel indirect effect. I do not find this to be clearly distinguishable (lines 393 to 415). The same reasoning applies for similar results.

It is customary to see in this line of literature that the equation of decomposition (first order derivative) and then report the direct effect, channel effect, total effect. It is customary to look into elasticities as follows: dgrowth/d(independent variable) = δgrowth/ δ(independent variable) + channel specific effect applied properly using the first derivative.

8-     The R-squared in Tables D6, D9, D12, …, and other tables is the same across different models (OLS, SUR, 2SLS, 3SLS) which is highly unlikely and is a concern.

Note: In a response to the authors response regarding R-squared, I note that this measure should not be interpreted as a goodness of fit measure in SLS type models.

9-     The authors should provide a descriptive statistics table which includes the number of observations used, and comment on whether the panel is balanced or not in the FE model.

The following points are made to enhance presentation:

a-     It is better to write about democracy and growth without referring to equations/variables in the first paragraph of the introduction since equations/variables are not yet introduced.

b-     Define all acronyms when first used.

c-     It is better to state “Several countries in the MENA region” rather than the MENA region on line 46 since lines 68 and 69 show a clear difference for Tunisia and Lebanon.

d-     Provide a reference for the claim made on lines 90/91.

e-     Add references for the claims made on lines 101/103.

f-      It is not necessary to note the violation of the OLS assumption referred to on line 290 since this is not the only assumption that is violated. Instead, I suggest that you state that we use 3SLS model to estimate … and then explain how the model addressed econometric issues in the area. It is also suggested to make a reference to other papers which used a close variation of the model.

g-     I assume that line 345 should read “after estimating the second stage”

h-     Line: 275 - Which classification is used by World bank? Provide a reference.

i-      It is better to consider using the previous point as the main reason for classifying countries as there exists significant heterogeneity across the states that make up the MENA region. This should be straightforward since you mention that the classification is equivalent to World Bank classification.

j-      Line 587: measures instead of measurements. 

Author Response

Dear Professor,

Kindly find attached below a file containing our responses to the required revisions of round two. We tried to implement them as much as possible.

Thank you very much.

Best Regards
